# A Polyclonal Antibody against a *Burkholderia cenocepacia* OmpA-like Protein Strongly Impairs *Pseudomonas aeruginosa* and *B. multivorans* Virulence

**DOI:** 10.3390/vaccines12020207

**Published:** 2024-02-17

**Authors:** António M. M. Seixas, Sara C. Gomes, Carolina Silva, Leonilde M. Moreira, Jorge H. Leitão, Sílvia A. Sousa

**Affiliations:** 1Department of Bioengineering, IBB—Institute for Bioengineering and Biosciences, Instituto Superior Técnico, Universidade de Lisboa, Av. Rovisco Pais, 1049-001 Lisboa, Portugal; antonio.seixas@tecnico.ulisboa.pt (A.M.M.S.); saracgomes@tecnico.ulisboa.pt (S.C.G.); carolina.sg.silva@gmail.com (C.S.); lmoreira@tecnico.ulisboa.pt (L.M.M.); 2Associate Laboratory, i4HB—Institute for Health and Bioeconomy, Instituto Superior Técnico, Universidade de Lisboa, Av. Rovisco Pais, 1049-001 Lisboa, Portugal

**Keywords:** *Pseudomonas aeruginosa*, *Burkholderia cenocepacia*, *Burkholderia multivorans*, cystic fibrosis, immunotherapy, antibody-based therapies, *Galleria mellonella*

## Abstract

Despite advances in therapies, bacterial chronic respiratory infections persist as life-threatening to patients suffering from cystic fibrosis (CF). *Pseudomonas aeruginosa* and bacteria of the *Burkholderia cepacia* complex are among the most difficult of these infections to treat, due to factors like their resistance to multiple antibiotics and ability to form biofilms. The lack of effective antimicrobial strategies prompted our search for alternative immunotherapies that can effectively control and reduce those infections among CF patients. Previous work from our group showed that the anti-BCAL2645 goat polyclonal antibody strongly inhibited *Burkholderia cenocepacia* to adhere and invade cultured epithelial cells. In this work, we showed that the polyclonal antibody anti-BCAL2645 also strongly inhibited the ability of *P. aeruginosa* to form biofilms, and to adhere and invade the human bronchial epithelial cell line CFBE41o-. The polyclonal antibody also inhibited, to a lesser extent, the ability of *B. multivorans* to adhere and invade the human bronchial epithelial cell line CFBE41o. We also show that the ability of *B. cenocepacia*, *P. aeruginosa* and *B. multivorans* to kill larvae of the *Galleria mellonella* model of infection was impaired when bacteria were incubated with the anti-BCAL2645 antibody prior to the infection. Our findings show that an antibody against BCAL2645 possesses a significant potential for the development of new immunotherapies against these three important bacterial species capable of causing devastating and often lethal infections among CF patients.

## 1. Introduction

Cystic fibrosis (CF) is a genetic recessive inherited disorder affecting an estimated 162,000 people worldwide, with 105,000 of these being diagnosed [1]. The incidence of CF is highly variable geographically. In Europe, for instance, this incidence ranges from 1 in 1400 live births to 1 in 25,000 in Ireland and Finland, respectively [2]. This disorder is due to mutations in the Cystic Fibrosis Transmembrane conductance Regulator gene (CFTR), leading to malfunctioning of the chloride channel [3]. This malfunctioning results in an increase in the mucus’s thickness in the lower respiratory tract and an impaired mucociliary clearance [3,4], creating a favorable environment for the colonization by various opportunistic pathogens. This colonization can progress to infection that frequently becomes chronic. The main pathogens of these chronic infections in CF patients are *P. aeruginosa* and bacteria of *Burkholderia cepacia* complex (Bcc), being *B. cenocepacia* and *B. multivorans* the more frequently isolated Bcc species [3,5]. Bcc are intrinsically resistant to the vast majority of clinically available antimicrobials [6,7]. This combination of antimicrobial resistance and the abundant variety of virulence factors produced by Bcc originates chronic infections with an unpredictable outcome, often lethal and nearly untreatable [8,9]. Currently, no effective strategies to eradicate Bcc bacteria from CF patients are available [10]. On the other hand, *P. aeruginosa* relies on adaptive mechanisms like switching from the planktonic to the biofilm mode of growth to cause chronic infections. Inside the biofilm, *P. aeruginosa* exhibit an increased tolerance to defense mechanisms of the host and resistance to antimicrobial therapy [11]. The importance of biofilm formation for the establishment of Bcc infections remains controversial, and some authors suggest that in the CF lung Bcc bacteria exist predominantly as single cells or in small clusters within phagocytes and in the mucus [12]. These chronic infections eventually induce respiratory failure, leading to death or lung transplantation [3]. Chronic infections by Bcc are characterized by a faster decline of pulmonary function coupled with exacerbation periods [13], with a significant percentage of patients developing cepacia syndrome—a rapid and fatal necrotizing pneumonia [14,15]. However, most of the morbidity and mortality among CF patients results from *P. aeruginosa* lung infections, with around 34% of CF adults from European countries experiencing chronic *P. aeruginosa* infections [3]. The establishment of chronic infections during the patient’s early age led to the introduction of intensive antimicrobial therapy, allowing numerous CF patients not to develop chronic infections during childhood [16]. This strategy, together with other therapeutic advances, increased the CF patients’ mean lifetime expectancy. CF patients are currently subject to intermittent or aggressive antibiotic treatment for management of chronic infections or during pulmonary exacerbations. The impact that these strategies have on the airway microbiota is unknown, with some studies suggesting that it is transient, with the bacterial communities recovering within 30 days [17,18]. However, the major drawback of these aggressive antibiotic therapies is a faster development of antimicrobial resistance and the lack of efficient therapies during exacerbation periods [17], highlighting the urgent need for new alternatives to fight these infections. Antibody-based immunotherapies allow a very specific activity without the risk of developing resistance in bacteria [19]. These strategies act by an assortment of mechanisms, including binding to antigens and preventing adherence and inhibiting other important steps of infection, or by mechanisms dependent on mediators, like antibody-mediated cellular cytotoxicity, complement-dependent cytotoxicity, or opsonization [19]. Several studies are currently being performed with potential new targets for the development of these new strategies [20]. In this work, we describe the ability of the anti-BCAL2645 polyclonal goat antibody to strongly inhibit *P. aeruginosa* adherence and invasion of the human bronchial epithelial cell line CFBE41o-, and to form biofilms. We also show that the antibody negatively impacts the ability of the exopolysaccharide-producer *B. multivorans* to adhere and invade the epithelial cell line, and to form biofilms. The antibody was previously demonstrated to interfere with the adhesion and invasion process of *B. cenocepacia* to bronchial epithelial cells [21]. The anti-BCAL2645 is here shown to recognize two homolog proteins from the clinical isolate *P. aeruginosa* F69A (isolate IST27) with its genome sequenced. The antibody is also shown to strongly decrease the virulence of the tested strains in the *Galleria mellonella* infection model. In conclusion, our findings highlight the potential of BCAL2645 as a viable candidate for the advancement of research on passive antibody-based therapies targeting both Bcc and *P. aeruginosa*.

## 2. Materials and Methods

### 2.1. Bacterial Strains and Culture Conditions

*P. aeruginosa* F69A isolate IST27 [22], *B. cenocepacia* K56-2 [23] and *B. multivorans* BM1 [24] were used in this work and maintained in PIA (Pseudomonas Isolation Agar plates; Becton Dickinson, Heidelberg, Germany). When in use, *Escherichia coli* was maintained in LB plates [21]. When necessary, media were supplemented with 150 μg/mL ampicillin. Unless otherwise stated, bacterial cultures were carried out at 37 °C in shake flasks with orbital agitation (250 rev/min) in liquid LB supplemented with adequate antibiotics. Bacterial growth was monitored spectrophotometrically at 640 nm (OD_640_).

### 2.2. Protein Expression and Purification

The DNA fragments encoding the *P. aeruginosa* genes *IPC84_RS02910* and of *IPC84_RS01840* were amplified by polymerase chain reaction (PCR), using the primer pairs UP-RS02910/LW-RS02910 (GTTCATATGTTCACCTCCCGTTG/ATTCTCGAGGTACTGCGGCTG), and UP-RS01840/LW-RS01840 (ATTCATATGAGCATCACGAGGA/TCTCTCGAGTTCGCGCTTGA), respectively. The PCR products were inserted into a pET23a(+) vector (Novagen, Madison, WI, USA) using a double digestion (NdeI/XhoI) method, creating the plasmids pCGS3 and pCGS4, respectively. The plasmids were transformed into *E. coli* BL21 (DE3) competent cells for overexpression of IPC84_RS02910 and IPC84_RS01840 proteins. The overexpression of both proteins was achieved upon culture induction with 0.4 mM IPTG at 30 °C. The 6× His-tagged proteins were purified by nickel affinity chromatography with a HisTrap FF column (GE Healthcare, Chicago, IL, USA). Increasing imidazole concentrations of 40, 60, 100, 150, 200, and 500 mM were used to remove impurities and elute the target proteins. Aliquots of 1 mL were collected for each elution concentration and analyzed by 15% SDS-PAGE.

### 2.3. Cell Line and Cell Culture

The CFBE41o-human bronchial epithelial cell line [25], was used in this study. Cells were cultured under humidified atmosphere at 37 °C with 5% CO_2_, as previously described [21].

### 2.4. Polyclonal Goat Antibodies

The anti-BCAL2645 antibody was obtained as described in Seixas et al. [21]. The anti-Hfq antibody was obtained from SICGEN (Portugal). The antibody was purified by immunoaffinity, and its specificity tested by Western blot against total cell extracts of *B. cenocepacia* J2315.

### 2.5. Western Blot Analyses

Proteins were analyzed using 12.5% (*w*/*v*) SDS-PAGE polyacrylamide gels. After electrophoresis, the proteins were transferred to nitrocellulose (NC) membranes (PALL corporation, New York, NY, USA) using a Trans-Blot^®^ SD (BIORAD, Hercules, CA, USA) device. The membranes were subsequently blocked overnight at a temperature of 4 °C using a 3% (*w*/*v*) BSA solution in PBS 1×. Western blot and chemiluminescence signal detection were performed as described previously [21].

### 2.6. Adhesion and Invasion of Epithelial Cells

Bacterial adhesion and invasion experiments were conducted using established methodologies as outlined in previous studies [21,26,27]. The bacterial strains were grown until an OD_640_ of 0.5 was reached. Subsequently, they were used to infect the 24 h cultivated cells, after adequate dilutions. The multiplicity of infection (MOI) used was 20:1 for *P. aeruginosa* and 50:1 for *B. multivorans*. To evaluate the suppressive impact of the antibody on the adhesion and invasion, the bacterial suspensions were subjected to a 1 h incubation period at room temperature, in the presence of the anti-BCAL2645 polyclonal goat antibody at a concentration of 50 µg/mL, prior to infection. To confirm the antibody specificity, an unrelated anti-Hfq polyclonal goat antibody was used at the same concentration. When assessing bacterial adhesion, cell monolayers were incubated for 30 min under 5% CO_2_ atmosphere at 37 °C, and then the wells were subsequently rinsed three times with PBS 1×. Then, 1 mL of lysis buffer (containing 10 mM EDTA and 0.25% (*v*/*v*) Triton X-100) was added. The cells were then incubated for an additional 20 min at 4 °C. To determine the number of bacteria adhering to the surface, the lysate was serially diluted and plated onto LB agar plates. In invasion assays, cells monolayers were incubated during 3 h for *P. aeruginosa* and 2 h for *B. multivorans*, allowing the bacteria to invade the cells. The wells were then rinsed 3× with PBS 1×, and incubated for an additional 2 h with MEM supplemented with 300 µg/mL gentamicin for *P. aeruginosa* and 1 mg/mL amikacin and 1 mg/mL ceftazidime for *B. multivorans*. Following this period, the effectiveness of the antibiotic treatment was verified by plating the supernatants. Wells were subsequently washed 3× with PBS 1× and incubated with 1 mL of lysis buffer for an additional period of 30 min at 4 °C. Following this procedure, serial dilutions were plated onto the surface of LB plates. Results were expressed as a ratio of the antibody-treated and untreated bacteria to adhere and invade epithelial cells, corrected with the initial bacterial dose applied.

### 2.7. Planktonic Cells Aggregation Assay

Planktonic cells aggregation assays were carried out based on previously described methods [28], with slight modifications. For this purpose, cultures of *P. aeruginosa* IST27 were grown overnight and diluted into Jensen’s medium to an initial OD_600_ of 0.01. Polyclonal goat antibodies were added at a concentration of 20 µg/mL and the cultures were incubated for 2 h at 37 °C. For the final 15 min, Congo Red (Sigma, St. Louis, MO, USA) was added (40 µg/mL) and the cultures were incubated for the remaining time. The culture density was measured at an OD_600_. 1 mL sample was removed and centrifuged at 20,000× *g* for 10 min. The remaining Congo Red in solution was determined by measuring the absorbance at 490 nm (A_490_). Finally, the ratio A_490_/A_600_ was calculated, and the data was plotted as a ratio of the value obtained for the WT strain. The results are expressed as a ratio, comparing the aggregation without antibody.

### 2.8. Biofilm Formation Assays

Biofilm formation was assessed using the crystal violet dye method [21,29]. The anti-BCAL2645 and Hfq antibodies were used at a concentration of 20 µg/mL. Results are mean values of at least 12 replicates from three independent experiments and are expressed as a ratio, comparing the biofilm formed by bacteria treated or not with antibody.

### 2.9. Galleria mellonella Killing Assays

*Galleria mellonella* killing assays were carried out based on methods already described [30,31] with slight modifications. After bacterial growth, bacterial cultures were measured at OD_640_, and the appropriate volume was collected and washed with PBS 1×. To assess if the antibody has a specific effect of inhibiting the killing of *G. mellonella*, prior to the infection bacterial suspensions (2 × 10^5^) were incubated with 50 µg/mL for Bcc bacteria and 25 µg/mL for *P. aeruginosa* of either the anti-BCAL2645 antibody or a control antibody for 1 h at room temperature. Dilutions were performed to obtain the number of bacteria per injection as follows: 1 × 10^2^ for *B. cenocepacia* K56-2 and *P. aeruginosa* IST27, and 2.5 × 10^4^ for *B. multivorans* BM1. The bacterial numbers used were previously optimized and confirmed by plating serial dilutions on LB agar plates. Bacterial injection was carried out as previously described. A total of 10 larvae were used for each condition assessed. The anti-Hfq antibody suspended in PBS (50 µg/mL) was used as the control. Antibody specificity was confirmed using an unrelated anti-Hfq polyclonal goat antibody at the same concentration. After injection, larvae were placed in Petri dishes and stored in the dark at 37 °C. Survival was monitored for 3 days for *B. cenocepacia* and *P. aeruginosa* and 5 days for *B. multivorans* at 24 h intervals. Larvae lacking movement upon touch were considered as death. *G. mellonella* survival was represented as Kaplan–Meier curves with data from the minimum of three independent experiments. The Gehan-Breslow-Wilcoxon statistical test was used to estimate differences in survival rates. The GraphPad Prism software, version 8.0.1, was used.

### 2.10. Protein Bioinformatics Analyses

The bioinformatics tools available at the National Center for Biotechnology Information (NCBI) [32] were utilized to analyze the nucleotide and predicted amino acid sequences. The conservation of protein domains across homolog proteins was assessed through the ExPASy-Prosite [33]. The *Burkholderia* Genome Database [34] and Pseudomonas Genome Database [35] were utilized to conduct searches for homologous sequences. Amino acid sequence alignments were performed with the help of Clustal Omega (EMBL-EBI) [36]. B-cell epitopes prediction was performed using BepiPred-2.0, from the Immune Epitope DataBase (IEDB), using a 0.5 threshold [37].

### 2.11. Statistical Analysis

The mean values of a minimum of three independent experiments were calculated and expressed as results, along with the corresponding standard deviations (SD). Statistical analysis was conducted using GraphPad Prism software (version 8.0). To determine statistically significant differences, both two-way and one-way analyses of variance (ANOVA) were performed. Results with a *p* value less than 0.05 were considered being statistically significant.

## 3. Results

### 3.1. Identification of BCAL2645 Homolog Proteins in the B. multivorans and P. aeruginosa Genomes

Previous work from our research group showed the immunological potential of the *B. cenocepacia* J2315 BCAL2645 protein for the development of immunotherapies [21,38]. In this study, we investigated the presence of BCAL2645 homologs in the genome of *P. aeruginosa*, aiming at the development of a potential immunotherapy able to protect from infections by both Bcc bacteria and *P. aeruginosa*. A BLASTP search allowed the identification of a protein 95.81% identical to BCAL2645 in the genome of *B. multivorans*, and of two possible homologs in the genome of *P. aeruginosa* F69A isolate IST27, a CF clinical isolate with its genome sequenced. The two *P. aeruginosa* identified proteins, IPC84_RS02910 and IPC84_RS01840, share, respectively, 55% and 35% identity to BCAL2645. An alignment of the amino acid sequences of the *P. aeruginosa* proteins with the BCAL2645 amino acid sequence is shown in Figure 1a,b. All the proteins under analysis exhibit the same domains, namely the Prokaryotic membrane lipoprotein lipid attachment site domain (PS51257) and the OmpA-like domain (PS51123), indicating they are all OmpA-like lipoproteins. Analysis of the predicted B-cell epitopes suggests that the epitopes within these proteins align. These results confirm the homology that exists between BCAL2645 protein and these two *P. aeruginosa* IST27 proteins. Next, to understand if the polyclonal antibody anti-BCAL2645 had cross-reactivity with proteins of other Bcc strains and *P. aeruginosa*, a Western blot against the whole proteins of both *P. aeruginosa* IST27 and *B. multivorans* BM1 (Figure 1c) was performed. It was observed a strong band with equivalent MW to BCAL2645 and 3 bands in the IST27 strain, being the higher and stronger band with around 25 kDa. This led us to purify the two proteins with higher identity, IPC84_RS02910 and IPC84_RS01840, to confirm if the anti-BCAL2645 antibody could recognize specifically these proteins. Our results (Figure 1d), show a significant reactivity against both proteins, being stronger with the IPC84_RS02910.

### 3.2. The Anti-BCAL2645 Polyclonal Antibody Interferes with P. aeruginosa Ability to Form Biofilms

The presence of the OmpA domain in the *P. aeruginosa* proteins IPC84_RS02910 and IPC84_RS01840 prompted us to investigate the effect of the anti-BCAL2645 antibody on the ability of the strain to produce biofilms. OmpA-like proteins have been previously shown to be important for biofilm formation and adhesion to several surfaces [21,39]. The incubation of *P. aeruginosa* IST27 in the presence of the anti-BCAL2645 antibody led to a reduction of about 30% and 40% of the biofilm formed after 24 h (Figure 2a) and 48 h (Figure 2b), respectively. As a control, we incubated the strains with a polyclonal antibody anti-HFQ, acquired from the same commercial company and produced using the same methods used for the anti-BCAL2645 antibody. To address whether the anti-BCAL2645 antibody interferes with cell–cell interactions and early aggregation events important for the irreversible attachment step in biofilm formation [40], *P. aeruginosa* cell–cell aggregation was investigated in the absence or presence of the anti-BCAL2645 antibody or the control anti-Hfq antibody. A reduction of about 20% in cell–cell aggregation was observed when cells were incubated in the presence of the anti-BCAL2645 antibody compared to the control antibody. Overall, these data suggest that the pAb anti-BCAL2645 is able to reduce the attachment of *P. aeruginosa* to biotic surfaces, and hindering this bacteria ability to form thicker biofilms.

### 3.3. The Anti-BCAL2645 Antibody Strongly Inhibits Adhesion and Invasion of Cultured Human Bronchial Epithelial Cells

We then investigated whether the anti-BCAL2645 pAB prevented *P. aeruginosa* and *B. multivorans* to adhere and invade the CFBE41o- human CF bronchial epithelial cell line. *P. aeruginosa* adhesion to cultured CFBE41o- cells was reduced by almost 80% when the bacteria were incubated with the anti-BCAL2645 antibody at a concentration of 50 μg/mL (Figure 3a). No significant differences were observed in the *P. aeruginosa* adhesion to epithelial cells when bacteria were pre-incubated with the anti-Hfq control antibody (Figure 3a). For the invasion process, the incubation of *P. aeruginosa* with the anti-BCAL2645 pAB led to a reduction of almost 80% of the total number of invading bacteria as compared to the anti-Hfq control antibody (Figure 3b). With *B. multivorans*, known to be an intracellular pathogen [41], incubation with the anti-BCAL2645 antibody led to a more modest reduction in the adhesion and invasion of the epithelial cells (Figure 3c,d). The *B. multivorans* BM1 strain produces high amounts of exopolysaccharide that might interfere with the anti-BCAL2645 inhibitory effects on adhesion and invasion of epithelial cells. In fact, previous work using the *B. cenocepacia* J2315 strain unable to produce exopolysaccharide, a stronger inhibitory effect of anti-BCAL2645 epithelial cell adhesion and invasion was reported [21]. The incubation of both *P. aeruginosa* and *B. multivorans* with the anti-Hfq antibody did not significantly affect the adhesion and invasion ability of both strains (Figure 3c,d).

### 3.4. The Anti-BCAL2645 Antibody Protects Galleria mellonella Larvae from B. cenocepacia, B. multivorans and P. aeruginosa Infections

The protective effect of the anti-BCAL2645 antibody against infection by *P. aeruginosa* and the different *Burkholderia* strains was assessed using the animal model *Galleria mellonella*. Prior to injection in the larvae, bacterial strains were incubated with the anti-BCAL2645 pAb. The pre-incubation of bacteria with the antibody led to a 90% survival of the larvae infected with *B. cenocepacia* K56-2 after 3 days (Figure 4a).

For *P. aeruginosa* the pre-incubation with the anti-BCAL2645 pAb led to a survival of 50% of the larvae (Figure 4b). With the *B. multivorans*, pre-incubation of bacteria with the antibody led to more modest effects, with only 39% of the infected larvae surviving after 3 days of infection (Figure 4c). Due to the lower virulence of the *B. multivorans* strain to the *G. mellonella* larvae, the assay was prolonged by 5 days. Nevertheless, in the control assay carried out with bacteria without pre-incubation with the antibody, a small percentage of larvae were still alive. Despite the lower protective effect of the antibody in this case, it is possible to observe a delay in the death of the larvae injected with bacteria previously incubated with the anti-BCAL2645 pAb (Figure 4c). Bacteria incubated with the anti-HFQ exhibited an ability to kill *G. mellonella* with an efficacy similar to that of bacteria untreated with anti-BCAL2645. Altogether, results show that the anti-BCAL2645 antibody is able to confer protection against infection across the species *B. cenocepacia*, *B. multivorans*, and *P. aeruginosa*.

## 4. Discussion

*P. aeruginosa* and Bcc respiratory infections among CF patients remain difficult to eradicate due to the ability of these bacteria to overcome antimicrobial therapies. This ability is related to the intrinsic resistance in Bcc, and with *P. aeruginosa*, to the antibiotic tolerance conferred by the biofilms formed by the bacterium. The exploitation of immunogenic surface-exposed proteins of pathogens to raise antibodies has the potential to generate novel therapeutics to surpass the growing inefficacy of antibiotics. Research on pathogen-specific antibodies is a fast-developing area, with large interest for infections caused by antimicrobial resistant bacteria [19]. Some studies using antibodies against *P. aeruginosa* have exhibited significant potential in preclinical infection models, although none are currently available [20]. For Bcc, the number of studied antibodies is significantly scarcer. Nonetheless, antibodies to specific pathogens are being regarded as one of the best alternatives to deal with multi-drug resistant infections [20]. OmpA-like proteins have been exploited as targets for the development of immunotherapies in several pathogens like *P. aeruginosa*, *Salmonella* spp., *Mannheimia haemolytica* and successfully used for the development of vaccines against Lyme disease [39,42]. This family of proteins play important roles in pathogenesis, being associated with biofilm formation, stimulation of proinflammatory cytokines, bacterial adhesion, invasion and intracellular survival [39,43,44,45,46]. In a previous work, we have shown that the anti-BCAL2645 pAb strongly inhibits the binding and invasion of epithelial cells by *B. cenocepacia* [21]. Here, we extended those studies to two additional CF pathogens, *P. aeruginosa* and the exopolysaccharide producer *B. multivorans.* Results presented here confirm the anti-BCAL2645 pAb ability to confer protection against these pathogens by inhibiting their adhesion and invasion of epithelial cells, reduce the formation of biofilms and the virulence towards the animal model *G. mellonella.*

Our first approach was the bioinformatics searches for protein similar to BCAL2645 in the genome sequences of *P. aeruginosa* and *B. multivorans*. Two *P. aeruginosa* and one *B. multivorans* proteins presented a significant identity to BCAL2645. More importantly, all the proteins displayed the Prokaryotic membrane lipoprotein lipid attachment site domain (PS51257) and the OmpA-like domain (PS51123). In addition, the predicted B-cell epitopes for the BCAL2645 and the homolog proteins seem to align and be located in regions of high similarity. These results indicating a high similarity of the common epitopes were confirmed by Western blot using the purified proteins, demonstrating that the pAB antiBCAL2645 recognizes the one *B. multivorans* and two *P. aeruginosa* proteins homolog to BCAL2645.

In *P. aeruginosa*, biofilms are critical for the development of persistent and chronic infections in CF patients [47]. As such, strategies able to diminish the ability of the bacterium to form biofilms are foreseen as highly efficient in preventing or, in combination with antimicrobials, to eradicate these infections. This is the case of the pAB anti-BCAL2645 that strongly inhibited the *P. aeruginosa* ability to form thick biofilms.

Adherence to host cells plays a vital role in the initiation and establishment of infections by both *P. aeruginosa* and Bcc. In *P. aeruginosa,* the early stages of infection are highly associated with the type IV pili that acts as an adhesin to bind to a variety of host cells [48]. *P. aeruginosa* is considered an extracellular pathogen, nonetheless, it can invade host cells, resulting in the internalization of these bacteria [48,49]. The invasion process is also associated with type IV pili [48,49]. Intracellular invasion by these bacteria is used as a strategic mechanism to elude the host immune system and to resist to antimicrobial therapy, facilitating the persistence and recurrence of infections [50]. In the specific case of CF patients, the CFTR plays a role as an epithelial cell receptor for *P. aeruginosa* [51]. Epithelial cells that possess a mutant form of the CFTR protein exhibit a significantly reduced internalization of *P. aeruginosa* compared to epithelial cells that express the normal CFTR protein [51]. As such, the invasion process by *P. aeruginosa* is less relevant in the CF infection context than with Bcc. Host cell invasion is a more significant process in the infection of the human cornea by *P. aeruginosa*, usually resulting in ulcerative keratitis, a rapidly progressing inflammatory response to bacterial infection of the cornea [49,51]. Although the pAb anti-BCAL2645 does not target the type IV pili, a highly significant reduction of the adhesion and invasion of the CFBE41o-human CF bronchial epithelial cell line was recorded when bacteria were previously incubated with the pAb. These results suggest that the specific binding of the antibody to the bacterium is causing a steric hindrance, obstructing the process of infection, hindering the initial steps for the establishment of infection. For Bcc bacteria, the process involved in the adherence to the epithelium is fairly studied but not yet fully understood, and some of the most studied effectors associated with adherence are not present in some of the most relevant clinical strains [52]. The ability of Bcc strains to invade lung epithelial cells is well established [53], however, the mechanisms underlining this process are not fully elucidated. The effect of the pAb on the adhesion and invasion of host cells by *B. cenocepacia* was previously shown [21]. However, the exopolysaccharide produced by the *B. multivorans* strain tested in this work most probably had a negative impact on the efficacy of the antibody by covering its target. We speculate that this effect is the main reason for the differences observed in the effects of the pAB anti-BCAL2645 on *B. cenocepacia* and *B. multivorans* ability to adhere and colonize the epithelial cells, and to kill the *G. mellonella* larvae. It is important to note that our results strongly indicate the anti-BCAL2645 antibody has potential as a new therapeutic antibody. However, polyclonal antibodies also have the potential to present major drawbacks, such as a high probability of inducing exacerbated host immune system responses. As such, further studies using monoclonal or IgY antibodies targeting BCAL2645 should be pursued in experimental infection models. In particular, the pharmacokinetics and pharmacodynamics of such anti-BCAL2645 antibodies and their ability to neutralize bacteria in a mammal infection model should be addressed. It is also worth mentioning that monoclonal or IgY antibodies have the potential of being used in antibody-antimicrobial combined therapies, or in combination with CF modulators such as elexacaftor, tezacaftor, and ivacaftor.

## 5. Conclusions

In conclusion, the anti-BCAL2645 pAB targeting the *B. cenocepacia* BCAL2645 is here demonstrated to specifically recognize proteins homolog to BCAL2645 from *B. multivorans* and *P. aeruginosa*. This specific recognition allows the pAb to interfere with various infection steps of these bacteria, by reducing the ability of *P. aeruginosa* to form biofilms and to adhere and invade human epithelial cells, two of the more important initial infection steps for these bacteria. The anti-BCAL2645 pAb also conferred a protective effect of the animal model *G. mellonella* when infected with the three tested bacterial strains were pre-incubated with the antibody. Altogether, our results show that an anti-BCAL2645 antibody has a high potential for the development of new immunotherapies against the most problematic strains infecting CF patients, *P. aeruginosa,* and Bcc.

## Figures and Tables

**Figure 1 vaccines-12-00207-f001:**
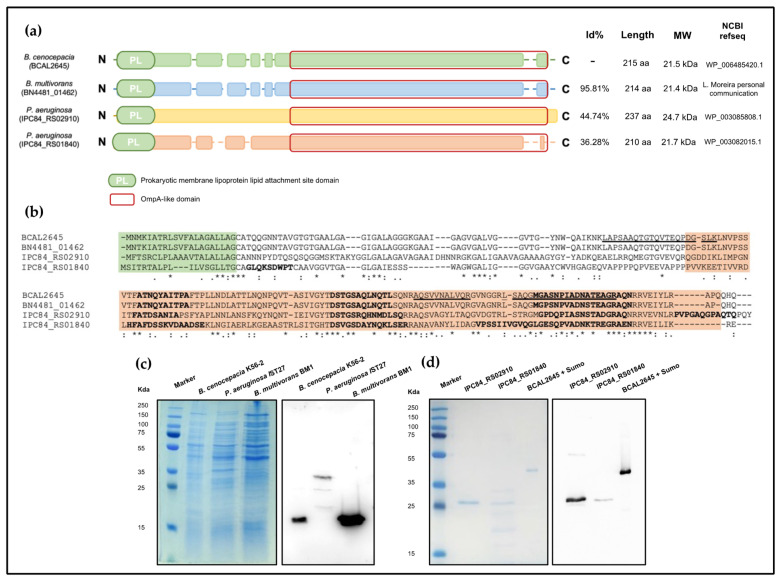
Alignment of amino acid sequences of *B. cenocepacia* J2315 BCAL2645 protein with the homologs IPC84_RS02910 and IPC84_RS01840 from *P. aeruginosa* IST27 and BN4481-01462 from *B. multivorans* BM1. (**a**) Analysis of BCAL2645 homolog proteins, showing the conserved domains and sequence identity (Id%). The length, molecular weight (MW) and National Center for Biotechnology Information (NCBI) reference sequence are also showed for the 4 proteins. (**b**) Amino acid sequence alignment of *B. cenocepacia* J2315 BCAL2645 protein, *P. aeruginosa* IST27 IPC84_RS02910 and IPC84_RS01840 proteins, and *B. multivorans* BM1 BN4481_01462. Bioinformatic predictions of B-cell epitopes are marked in bold. Prokaryotic membrane lipoprotein lipid attachment site domain (PS51257) and OmpA-like domain (PS51123) are highlighted in green and orange, respectively. The BCAL2645 surface-exposed peptides previously identified by Sousa et al. [38] are underlined. Asterisks indicate identical amino acid residues, one or two dots indicate semi-conserved or conserved substitutions, respectively. (**c**) SDS-PAGE and Western blot analysis using the polyclonal anti-BCAL2645 antibody and the whole proteins of *B. cenocepacia* K56-2, *P. aeruginosa* IST27 and *B. multivorans* BM1 (**d**) SDS-PAGE analysis of the purified recombinant proteins and Western blot analysis using the polyclonal anti-BCAL2645 antibody against the purified recombinant proteins IPC84_RS02910 and IPC84_RS01840 of *P. aeruginosa* IST27. The BCAL2645 protein, overexpressed with a N-terminal SUMO fusion, was used as a positive control.

**Figure 2 vaccines-12-00207-f002:**
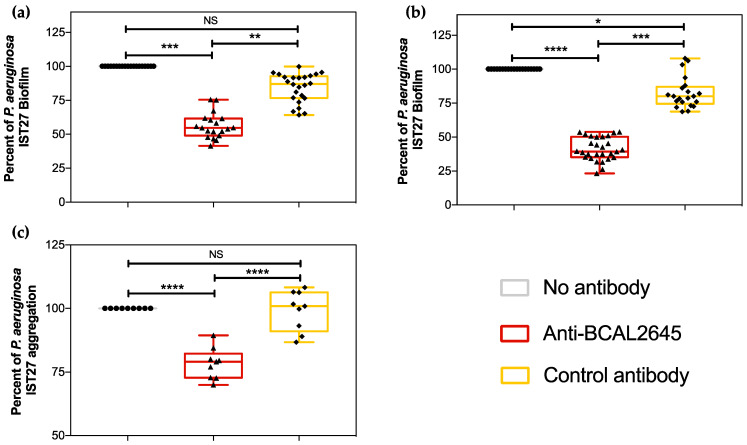
The anti-BCAL2645 antibody reduces cell–cell aggregation in *P. aeruginosa* and its ability to form thicker biofilms. (**a**,**b**) Anti-BCAL2645 pAb (20 µg/mL) inhibits *P. aeruginosa* IST27 biofilm formation at 24 and 48 h, respectively, and (**c**) the cell–cell aggregation in liquid culture. Results were expressed as percentage of biofilm and aggregation relatively to biofilm and aggregation of *P. aeruginosa* IST27 strain with no treatment. The pAb anti-Hfq was used as the control antibody. Statistics were determined using by one-way ANOVA. All the results are presented as the mean from three independent experiments, error bars indicate SD. (****, *p* < 0.0001; ***, *p* < 0.001, **, *p* < 0.01; * *p* < 0.05; NS, not significant).

**Figure 3 vaccines-12-00207-f003:**
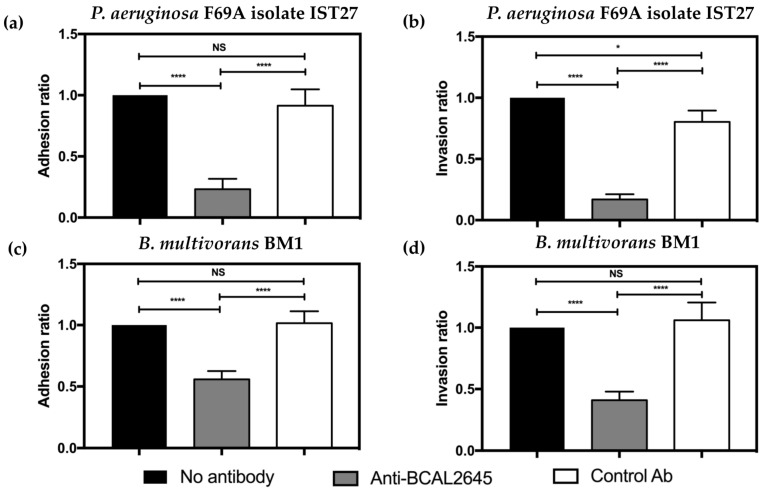
The anti-BCAL2645 antibody reduces host-cell adhesion and invasion abilities of both *P. aeruginosa* and *B. multivorans*. (**a**) Adherence inhibition of *P. aeruginosa* IST27 towards CF bronchial epithelial cell line 16HBE14o- after incubation with the anti-BCAL2645 antibody (50 μg/mL). (**b**) Invasion inhibition of *P. aeruginosa* IST27 towards CF bronchial epithelial cell line 16HBE14o- after incubation with anti-BCAL2645 antibody (50 μg/mL). (**c**) Adherence inhibition of *B. multivorans* BM1 towards CF bronchial epithelial cell line 16HBE14o- after incubation with anti-BCAL2645 antibody (50 μg/mL). (**d**) Invasion inhibition of *B. multivorans* BM1 towards CF bronchial epithelial cell line 16HBE14o- after incubation with anti-BCAL2645 antibody (50 μg/mL). The antibody anti-HFQ was used as a control in adhesion and invasion experiments at a concentration of 50 μg/mL. Results were expressed as percentage of adhesion and invasion relative to adhesion and invasion of the strain with no treatment. Results are presented as the mean of three independent experiments. Statistic analysis were performed by one-way ANOVA. All the results are presented as the mean from three independent experiments, error bars indicate SD. (****, *p* < 0.0001; *, *p* < 0.05; NS, not significant).

**Figure 4 vaccines-12-00207-f004:**
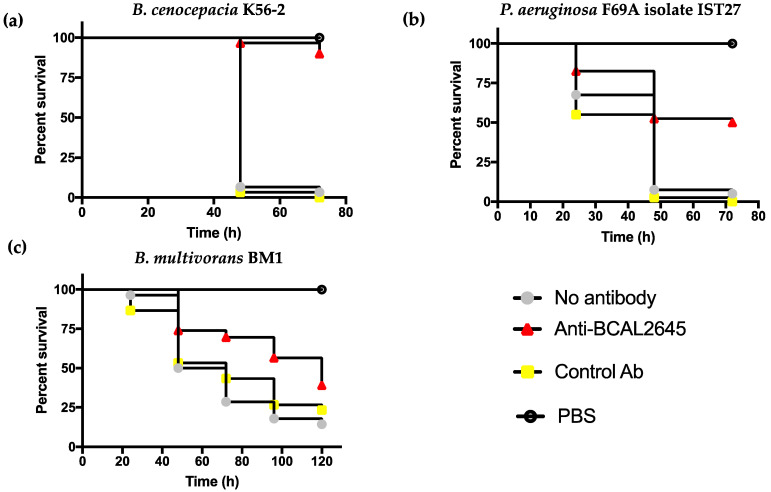
The pAb anti-BCAL2645 protects larvae of *Galleria mellonella* against bacterial infection caused by *B. cenocepacia*, *B. multivorans* and *P. aeruginosa*. Kaplan–Meier graphs of *G. mellonella* survival after injection with *B. cenocepacia* K56-2 (**a**), *P. aeruginosa* IST27 (**b**), and *B. multivorans* BM1 (**c**). Bacteria were previously incubated or not with anti-BCAL2645 or with the anti-HFQ control antibody. Uninfected larvae injected with PBS were also used as control. Results represent the mean of three independent determinations for 10 animals per treatment. Statistics were determined using Gehan-Breslow-Wilcoxon test. For *B. cenocepacia* K56-2 a *p* value < 0.0001 was obtained compared to no Ab and control Ab. For *P. aeruginosa* IST27 a *p* value < 0.001 was obtained compared to no Ab and *p* value < 0.0001 against control Ab. For *B. multivorans* BM1 a *p* value < 0.01 was obtained comparing against no Ab and *p* value < 0.05 against control Ab.

## Data Availability

The data presented in this study are available on request from the corresponding author.

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
