# Peer review of "A Polyclonal Antibody against a Burkholderia cenocepacia OmpA-like Protein Strongly Impairs Pseudomonas aeruginosa and B. multivorans Virulence"

_vaccines, 2024, doi:10.3390/vaccines12020207_

Round 1

Reviewer 1 Report

Comments and Suggestions for Authors

This paper described effects of a polyclonal goat antiseraon the ability of a variety of microorganisms involved in respiratory complications in cystic fibrosis to interact with epithelial cell lines, in essence. Design and data are correctly managed. I, particularly, miss identification of the molecules targeted and/or involved in the phenomenon. Otherwise results ar of sound an conclusions well drawn.

Author Response

REVIEWER 1

“This paper described effects of a polyclonal goat antisera on the ability of a variety of microorganisms involved in respiratory complications in cystic fibrosis to interact with epithelial cell lines, in essence. Design and data are correctly managed. I, particularly, miss identification of the molecules targeted and/or involved in the phenomenon. Otherwise, results are of sound and conclusions well drawn.”

Authors Answer:

We kindly appreciate the general comments made by the reviewer. The molecules targeted by the antibody are the BCAL2645 OmpA-like protein from B. cenocepacia and their homologues from B. multivorans and P. aeruginosa, as demonstrated by the results present in the whole section 3.1. - Identification of BCAL2645 homolog proteins in the B. multivorans and P. aeruginosa Genomes. Results from western blots against whole proteins from the three organisms indicate that only signals corresponding to BCAL2645 homologs were detected. Further explanations are given in the text of section 3.1.

Reviewer 2 Report

Comments and Suggestions for Authors

António M.M. Seixas and colleagues performed in vitro investigation to assess if a polyclonal antibody J2315 BCAL2645 against OmpA-like protein of Burkholderia cenocepacia shows cross-interaction with homologous proteins from Pseudomonas aeruginosa and B. multivorans with potential clinical implications for cystic fibrosis patients.

The authors performed alignment of homologous proteins to find that they indeed share conserved domains. WB analyses confirmed binding of the  J2315 BCAL2645 to whole protein, lysates of P. aeruginosa and B.multivorans.

While binding of the antibody to the B. multivorans is convincing on the WB, it is not clear for P. aeruginosa.

Subsequently, the authors tested the impact of the antibody on biofilm formation showing significant reduction compared to a control antibody. anti-BCAL2645 proved to be effective also in reduction of host-cell adhesion assay.

Finally, anti-BCAL2645 showed a protective effect on Galleria mellonella against infection with B. cenocepacia, B. multivorans and P. aeruginosa.

Question 1:

Can the authors explain why does the P. aeruginosa protein show higher on the WB compared to the others tested? Is the OmpA-like protein mass different for the 3 different bacteria?

Question 2:

Authors should indicate if the is a statistical difference between the control antibody and no treatment control in the biofilm assay just as they did for the host-cell adhesion assay.

Question 3:

Concentrations of anti-BCAL2645 differ between the biofilm and cell adhesion assay. Is the 20µg/mL concentration too low to observe an effect in adhesion reduction. Can the authors indicate what is the expected effective in-vitro concentration of the antibody and what levels of antibodies are generally considered safe for administration?

Question 4:

The authors should discuss the potential drawbacks of using a polyclonal antibody in a therapeutic setting or discuss how could it be further developed to be safe. How would the treatment be administered (inhaled or systemic?) I am missing discussion on potential off-target effects of the antibody and impact on the human microbiome is administered systematically. In a similar way, the introduction could benefit from a broader introduction of antibody treatments for infectious diseases. The need for such treatment in CF patients should be also discussed in the context of successful CF treatments with elexacaftor, tezacaftor and ivacaftor triple therapy.

Author Response

REVIEWER 2

António M.M. Seixas and colleagues performed in vitro investigation to assess if a polyclonal antibody J2315 BCAL2645 against OmpA-like protein of Burkholderia cenocepacia shows cross-interaction with homologous proteins from Pseudomonas aeruginosa and B. multivorans with potential clinical implications for cystic fibrosis patients.

The authors performed alignment of homologous proteins to find that they indeed share conserved domains. WB analyses confirmed binding of the J2315 BCAL2645 to whole protein, lysates of P. aeruginosa and B. multivorans.

While binding of the antibody to the B. multivorans is convincing on the WB, it is not clear for P. aeruginosa.

Subsequently, the authors tested the impact of the antibody on biofilm formation showing significant reduction compared to a control antibody. anti-BCAL2645 proved to be effective also in reduction of host-cell adhesion assay.

Finally, anti-BCAL2645 showed a protective effect on Galleria mellonella against infection with B. cenocepacia, B. multivorans and P. aeruginosa.

Question 1:

Can the authors explain why does the P. aeruginosa protein show higher on the WB compared to the others tested? Is the OmpA-like protein mass different for the 3 different bacteria?

Authors Answer:

Thanks for your kind appreciation of our work.

The molecular weights (MW) of the homolog proteins are described in Figure 1a, where is possible to observed that one of the homologs from P. aeruginosa, the IPC84_RS02910, has a higher molecular mass. The other 3 proteins have similar theoretical MW, however on SDS-PAGE gel the Burkholderia strains proteins run faster on gel compared with the Pseudomonas strain proteins. This difference on gel mobility could be due to different amino acid composition. In fact, anomalous gel mobility of membrane proteins, also known as “gel shifting” is a common feature described, being explained by Rath et al. (2009) (https://doi.org/10.1073/pnas.081316710) that is caused by alterations in detergent binding of these proteins.

Question 2:

Authors should indicate if the is a statistical difference between the control antibody and no treatment control in the biofilm assay just as they did for the host-cell adhesion assay.

Authors Answer:

Thanks for the observation. We have included the statistical difference between the control antibody and no treatment control in the new Figure 2. We have slightly modified the legend to accommodate the changes in new Figure 2.

Question 3:

Concentrations of anti-BCAL2645 differ between the biofilm and cell adhesion assay. Is the 20µg/mL concentration too low to observe an effect in adhesion reduction. Can the authors indicate what is the expected effective in-vitro concentration of the antibody and what levels of antibodies are generally considered safe for administration?

Authors Answer:

Thank for the observation. The concentration of antibody used was chosen according to the literature and after preliminary analysis of a range of different concentration for each different assay tested. For the adhesion assays, the concentration used was the lowest tested that led to a statistics significant difference comparing with the controls. For the biofilms, 20µg/mL was the concentration present in the literature for P. aeruginosa using monoclonal antibodies. Therefore, this was the only concentration tested. No concentration lower than 50µg/mL was tested for cell adhesion assays and for comparison between the different bacterial species the concentration used was maintained.

In vitro analysis of the antibody revealed an effective inhibition of biofilm formation, reduction of cells adhesion and lower virulence in Galleria melonella at concentrations between 20 µg/mL to 50 µg/mL. These concentrations are lower than the therapeutic doses already tested for other therapeutic antibodies. However, to consider an antibody safe for administration, it always depends on the specific molecule and their pharmacokinetic. In specific, monoclonal antibodies are often used at high doses, with therapeutic doses ranging between 5 mg and 750 mg per patient, with subcutaneous administration with a limited injectable volume of ~1–2 mL per dose, reaching concentrations often higher than 100 mg/mL (10.3390/molecules24142528). Since this work does not involve testing the antibody in mammals, we have not included this explanation in the text.

Question 4:

The authors should discuss the potential drawbacks of using a polyclonal antibody in a therapeutic setting or discuss how could it be further developed to be safe. How would the treatment be administered (inhaled or systemic?) I am missing discussion on potential off-target effects of the antibody and impact on the human microbiome is administered systematically. In a similar way, the introduction could benefit from a broader introduction of antibody treatments for infectious diseases. The need for such treatment in CF patients should be also discussed in the context of successful CF treatments with elexacaftor, tezacaftor and ivacaftor triple therapy.

Authors Answer:

Thanks for the suggestion. A short paragraph was added to the discussion section: “Its important to note that our results strongly indicate the anti-BCAL2645 antibody has potential as a new therapeutic antibody. However, polyclonal antibodies have also the potential to present major drawbacks, such as a high probability of inducing exacerbated host immune system responses. As such, further studies using monoclonal or IgY antibodies targeting BCAL2645 should be pursued in experimental infection models. In particular, the pharmacokinetics and pharmacodynamics of such anti-BCAL2645 antibodies and their ability to neutralize bacteria in a mammal infection model should be address. It is also worth to mention that monoclonal or IgY antibodies have the potential of being used in antibody-antimicrobial combined therapies, or in combination with CF modulators such as elexacaftor, tezacaftor and ivacaftor.”

We thank for the suggestion of including a “discussion on potential off-target effects of the antibody and impact on the human microbiome is administered systematically”. However, the discussion of such a topic is highly speculative based on the results presented, and we opted not to include such a discussion. Nevertheless, is worth to mention that the antibody tested always showed a high specificity, only recognizing the target protein and their homologs (Figure 1C). In addition, when used against E. coli, the antibody was unable to recognize any protein (our data not shown), but we haven´t tested any additional strains.

Reviewer 3 Report

Comments and Suggestions for Authors

Thank you for asking me to review this manuscript. It is very clearly written and I only have a few comments.

Line 38: The worldwide prevalence of Cystic Fibrosis is quoted as ~70,000. This is quoted from a reference dated to 2013. In fact, more recent figures suggest that the prevalence is more than double this, and the authors may wish to change this figure cited an up to date reference.

Line 46: Instead of using the phrase ’The main causers’ it would be better to state ‘ The pathogens in chronic infections in CF patients…’

Discussion: The data described is in vitro and in models. The big question is whether the neutralising capacity of antibody can be harnessed within patients. A short paragraph on this would be helpful to the readership in understanding the bridge to applicability. Why with repeated exposures to these pathogens do cystic fibrosis patients not already have sufficient neutralising antibodies? Is immunisation with key epitopes envisaged? If exogenous antibodies are administered would they reach the parts of the airways where they are needed and have a sustained effect? I appreciate that these questions may not yet have clear answers.

Comments on the Quality of English Language

/

Author Response

REVIEWER 3

Thank you for asking me to review this manuscript. It is very clearly written and I only have a few comments.

Line 38: The worldwide prevalence of Cystic Fibrosis is quoted as ~70,000. This is quoted from a reference dated to 2013. In fact, more recent figures suggest that the prevalence is more than double this, and the authors may wish to change this figure cited an up to date reference.

Authors Answer:

The authors kindly appreciate the comments of the reviewer. The worldwide prevalence of CF and the reference quoted was updated. The new sentence was included: “Cystic fibrosis (CF) is a genetic recessive inherited disorder affecting an estimated 162,000 people worldwide, with 105,000 of these being diagnosed [1].”

Line 46: Instead of using the phrase ’The main causers’ it would be better to state ‘ The pathogens in chronic infections in CF patients…’

Authors Answer:

Thank you for the observation. The expression “The main causers” was used to give emphasis that although several pathogens can cause chronical infection in CF, these are two of the most problematic.  Therefore, we have changed the text to “ The main pathogens in….”

Discussion: The data described is in vitro and in models. The big question is whether the neutralising capacity of antibody can be harnessed within patients. A short paragraph on this would be helpful to the readership in understanding the bridge to applicability. Why with repeated exposures to these pathogens do cystic fibrosis patients not already have sufficient neutralising antibodies? Is immunisation with key epitopes envisaged? If exogenous antibodies are administered would they reach the parts of the airways where they are needed and have a sustained effect? I appreciate that these questions may not yet have clear answers.

Authors Answer:

Thank you for the observation. Please see our response to Reviewer #2, question #4.

Reviewer 4 Report

Comments and Suggestions for Authors

I congratulate the authors on their work and a well-written manuscript.

Authors demonstrate that the capacity of  P. aeruginosa to adhere to and penetrate human bronchial epithelial cells, as well as to form biofilms, is significantly inhibited by polyclonal antibody anti-BCAL2645. Additionally, the polyclonal antibody partially prevents B. multivorans from adhering to and invading the human bronchial epithelial cell line.  The study revealed that the ability of B. cenocepacia, P. aeruginosa, and B. multivorans to kill larvae of the Galleria mellonella model of infection was reduced when these bacteria were incubated with an anti-BCAL2645 antibody before the infection. This significant finding suggests that the antibody against BCAL2645 has  potential for developing new immunotherapies against these three essential bacterial species that can cause severe and often fatal infections among CF patients.

Author Response

REVIEWER 4

 I congratulate the authors on their work and a well-written manuscript.

Authors demonstrate that the capacity of P. aeruginosa to adhere to and penetrate human bronchial epithelial cells, as well as to form biofilms, is significantly inhibited by polyclonal antibody anti-BCAL2645. Additionally, the polyclonal antibody partially prevents B. multivorans from adhering to and invading the human bronchial epithelial cell line.  The study revealed that the ability of B. cenocepacia, P. aeruginosa, and B. multivorans to kill larvae of the Galleria mellonella model of infection was reduced when these bacteria were incubated with an anti-BCAL2645 antibody before the infection. This significant finding suggests that the antibody against BCAL2645 has  potential for developing new immunotherapies against these three essential bacterial species that can cause severe and often fatal infections among CF patients.

Authors Answer:

The authors highly appreciated the general comments about the work.